# Engineering Transport via Collisional Noise: A Toolbox for Biology Systems

**DOI:** 10.3390/e26010020

**Published:** 2023-12-24

**Authors:** Alessandro Civolani, Vittoria Stanzione, Maria Luisa Chiofalo, Jorge Yago Malo

**Affiliations:** Dipartimento di Fisica Enrico Fermi, Università di Pisa and INFN, Largo B. Pontecorvo 3, I-56127 Pisa, Italy; alessandro.civolani7@gmail.com (A.C.); marilu.chiofalo@unipi.it (M.L.C.)

**Keywords:** open quantum systems, stochastic collision models, quantum transport in noisy media, quantum biology, quantum spin models

## Abstract

The study of noise assisted-transport in quantum systems is essential in a wide range of applications, from near-term NISQ devices to models for quantum biology. Here, we study a generalized XXZ model in the presence of stochastic collision noise, which allows describing environments beyond the standard Markovian formulation. Our analysis through the study of the local magnetization, the inverse participation ratio (IPR) or its generalization, and the inverse ergodicity ratio (IER) showed clear regimes, where the transport rate and coherence time could be controlled by the dissipation in a consistent manner. In addition, when considering various excitations, we characterized the interplay between collisions and system interactions, identifying regimes in which transport was counterintuitively enhanced when increasing the collision rate, even in the case of initially separated excitations. These results constitute an example of an essential building block for the understanding of quantum transport in structured noisy and warm-disordered environments.

## 1. Introduction

Novel technological developments in quantum mechanical systems have allowed including the effects of dissipative coupling to the environment, providing not only a more realistic characterization of the hardware [1,2,3], but also, with the current level of control and tunability, giving access to novel non-equilibrium physical phenomena not appearing in closed systems: from the engineering of new states of matter [4,5] or the observation and description of dissipative phase transitions [6,7] to the control of dynamical rates [8,9], with examples in discrete [10,11,12] and continuous systems [13,14], among many other applications. In addition, the study of transport in noisy media has become essential for its potential applications in the description of biological systems. Following the discovery of long-coherence times at room temperature in complexes involved in photosynthesis [15], only possible with the development of femtosecond two-dimensional spectroscopy methods [16,17,18,19], a large community has formed to question whether nature exploits the presence of coherent and dissipative couplings to further enhance efficiency [20,21].

In this context, we are interested in the phenomenological description of transport phenomena, with the use of the toolbox provided by open quantum systems. In particular, we focus on bath descriptions that allow for flexible couplings and architectures that and are compatible with noisy intermediate-scale quantum (NISQ) technology applications. As a result, we focus on the use of *quantum collision models* and *repeated interaction schemes* [22,23,24,25,26,27,28,29,30,31,32]. This discrete bath description consists of itinerant degrees of freedom (d.o.f.) that interact instantaneously and one at a time with the system or a part of it. As a result, this it suitable for describing discrete lattice system or quantum computing (QC) devices. Importantly, it also bears strong similarities with biological systems, where the collisions with ancillary systems can provide a good approximation of the noise present in those warm media. Thus, these models represent an efficient approach to building effective models for the description of such biological systems [33,34] that are also compatible with current NISQ devices for simulation.

Moreover, some of these applications have more recently been equipped with a toolbox of complex networks [35], creating a platform to study both the underlying microscopic transport phenomena, together with the emergence of macroscopic properties, and leading to the development of a generalized theory composed of both quantum and classical tools [36]. The study of these networks and their transport properties, in their classical and quantum version, is referred to as quantum walks [37]. Dissipative versions of these problems, more recently developed, are being exploited to also tackle problems relevant to biology [38,39].

Inspired by these lines of research, in this work, we consider a specific sub-branch of collision models, the *stochastic collision models* (SCM) [33,40,41,42,43]. In this formulation, the collisions are governed by a stochastic process and the unraveling of the evolution of the system can be described in terms of individual random realizations, which are tightly bound with individual runs on a given QC hardware. As our aim is to consider the most general noise distribution, we take inspiration from [33] and resort to the use of Weibull renewal processes [44] as our stochastic distribution for a general and tunable noise description. In fact, this distribution allows exploring a variety of bath-induced noise regimes, by tailoring both the rate of collisions over time and their space and time homogeneity. This model allows exploring non-trivial structured baths via stochastic sampling [33,40,45] with a simple theoretical description comparable to other stochastic unravelings in standard Markovian dynamics [4].

Furthermore, quantum spin models have been used as buildings blocks for the study of these natural networks [21] with architectures often based on the specific biological complex involved [33]. In our case, we instead consider an anisotropic linear chain, with the idea of reducing the impact of the network topology and isolating the role of the dissipative coupling in the transport. In particular, in this work, we investigate a paradigmatic quantum spin chain given by the Heisenberg XXZ model [46,47,48], motivated by both its applicability to describing phenomenology in other biological systems, such as neuroscience of perception [49], and also, given its integrability, its historical use in the study of transport [50,51]. Thus, in this work we combine this quantum spin model with the SCM to investigate the quantum transport of spin excitations and to also access the interplay between coherent and dissipative dynamics in the presence of anisotropy, leading to spin–spin interaction.

This paper is structured in the following manner: In Section 2, we present the spin model and the details of the noise implementation, together with the state of the art in the understanding of transport slowdown via dissipation. In Section 3, we present the transport analysis, while varying the noise parameters: we build an understanding for the case of one single excitation, before extending to several interacting excitations. From our analysis, we show that noise can be engineered to control the system transport properties. While normal structure-less noise leads to transport slowdown [13], we find specific regimes where we can minimize this effect, tailor the amount of coherent oscillations of the dynamics, or even enhance transport in the case where the excitations are pinned together due to the spin–spin interaction. Finally, in Section 4 we discuss our findings and potential future directions linked to disordered systems and complex networks.

## 2. Model and Numerical Method

Here, we consider an integrable generalization of the Heisenberg spin chain that accounts for uni-axial anisotropy in the spin interaction, generally on the *z*-axis, and described by the Hamiltonian [48]:(1)$$H_{XXZ}=J\sum_{i=1}^{N-1}\left[\sigma_{i}^{x}\sigma_{i+1}^{x}+\sigma_{i}^{y}\sigma_{i+1}^{y}+\Delta \sigma_{i}^{z}\sigma_{i+1}^{z}\right]+h\sum_{i=1}^{N}\sigma_{i}^{z},$$
with open boundary conditions (OBC). Here, *N* represents the number of spins, *J* is the exchange constant, anisotropy is represented by the parameter Δ, with *h* representing a generic transversal field. This is schematically shown in Figure 1. In the limit of moderate interaction Δ, the sign of the coupling *J* determines the ground state properties of the system, which develops a *ferromagnetic* (J>0) or *anti-ferromagnetic* (J<0) order. Instead, when considering time-evolution or quench dynamics, we can see *J* as the tunneling rate of an excitation or spin flip.

Unless stated otherwise, we work at fixed J=1 as our frequency unit and choose h=0, focusing on the case with Δ>0 and making use of the following system’s symmetry:(2)$$\left[H_{XXZ},S^{z}\right]=0,$$
meaning that the global magnetization along the *z*-axis, $S^{z}=\sum_{i}\sigma_{i}^{z}/2$, is preserved.

In order to incorporate the effects of noise, we introduce the environment via a discretization, using the so-called quantum collision models [52,53,54,55], see Figure 1a. In particular, we focus on one of their sub-branches: stochastic collision models [33,40,56,57]. In these models, the collisions are governed by a stochastic process, and the unraveling of the system’s evolution can be described in terms of single individual realizations. In this type of framework, it is possible to introduce any probability distribution to describe the collisional rate. Since we are interested in understanding the role of the noise parameters in transport and the time evolution of the system, we consider, as in [33], a flexible distribution given by a Weibull distribution [44]:(3)$$p\left(t\right)=\frac{\nu }{\mu }{\left(\frac{t}{\mu }\right)}^{\nu -1}e^{-{\left(t/\mu \right)}^{\nu }},$$
where ν≥0, the shape parameter, and μ>0, the scale parameter, are the collision parameters of the distribution. The shape parameter controls the temporal heterogeneity of the noise, (see Figure 1b), describing heterogeneous collisions over time for ν≤1 and temporal homogeneity for ν≫1, as the intercollision time becomes constant. In contrast, the scale parameter is related to the overall collision rate, which we define below:(4)$$r_{c}=\frac{1}{\tau_{th}}=\frac{1}{\mu \;\Gamma \;(1+1/\nu )},$$
with $\tau_{th}$ describing the mean collision time and Γ describing the Γ function obtained by evaluating $\int_{0}^{t}\tau p\left(\tau \right)d\tau =\Gamma \left(1+1/\nu \right)$ from Equation (Equation 3). As these rates can be tuned locally, changing the shape $\nu_{i}$ and scale $\mu_{i}$ parameters for each individual chain element can create not only temporal but also spatial heterogeneity in the system.

Now focusing on the evolution of the system, it is important to note that it is not possible to derive a GSKL-like master equation, as in the case of the Poisson distribution [40], for a Weibull distribution given by Equation (Equation 3). Despite this difficulty, it is still possible to describe the evolution in terms of quantum channels, see [33,40], and relying on stochastic unraveling.

To do so, we initialize a list $\bar{S}$ containing the first collision time of each site, randomly sampled from our distribution, and extract from it $S_{i}=\min\left(S_{j}\;|\;\forall \;j\in \left[1,N\right]\;\right)$, the shortest waiting time before a collision. We then evolve the density matrix by computing its evolution until the time of the first collision, i.e., $\rho \left(S_{i}\right)=e^{-iH(S_{i}-t)}\;\rho \left(t\right)\;e^{iH(S_{i}-t)}$. At this time, we apply the quantum channel
(5)$$\Phi \left[\rho (S_{i})\right]=\mathrm{Tr}\left[U_{\mathrm{coll}}\left(\rho_{a}⊗\rho \left(S_{i}\right)\right)U_{\mathrm{coll}}^{†}\right]\;,$$
with $U_{\mathrm{coll}}=\exp-i\left(\pi /2\right)\sigma_{a}^{x}⊗\sigma_{i}^{z}$ representing the collision event and $S_{i}$ being the shortest waiting time before a collision. Note that here we have chosen the collisional event to produce dephasing in the system—as this is proportional to $\sigma_{i}^{z}$ in the system. Thus, we are modeling a type of collision in which the interaction with the ancilla uncorrelates the system from other local d.o.f., while the same formalism can be adapted to other phenomena, e.g., collisions leading to the loss of the excitation ($\propto \;\sigma_{i}^{-}$).

After the computation of the first collision event, we can continue the process and redraw from the distribution in Equation (Equation 3) the next collision time for the *i*-th site, which will be $S_{i}=S_{i}+\tau$, with τ a random time sampled from the distribution; note that this would in general differ from its average, given by the mean collision time $\tau_{th}$. We then update $\bar{S}$ and extract the next collision time $S_{i}^{\prime }$, and distinguish two situations: if t+dt, with dt our numerical timestep, is larger than the new $S_{i}^{\prime }$, we repeat the corresponding application of the new quantum channel; otherwise, we proceed with the time evolution by simply computing $\rho \left(S_{i}^{\prime }\right)=e^{-iH\;(S_{i}^{\prime }-t)}\rho \left(S_{i}^{\prime }\right)e^{iH\;(S_{i}^{\prime }-t)}$ and then apply the quantum channel. By iteratively repeating this process, we can compute the time evolution of the system up to the desired time *T*.

## 3. Results

In this section, we present the results of our numerical analysis of the transport behavior and relevant observables quantifying it. Since we consider OBC, we will refer to the final time of the simulations $t_{f}$ as the last time step at which the spreading of the magnetization is computed, given by the time at which the magnetization probability differs from its initial value in the boundary sites to minimize the finite size effects. After our numerical convergence analysis (see Appendix A), we concluded that choosing dt=0.02 and the number of stochastic realizations M=500 provided us with reasonable results in every noise situation.

We start from an initial configuration with one (more) excitation(s), given by a flipped spin in a completely polarized state in the central site(s). Then, we analyze the rate of spreading of the magnetization as a function of the system and noise parameters. We expect to observe that the transport is generally slowed down when increasing the number of collisional events, as shown in Figure 1c, so that we consider the limits of ballistic ($r_{c}\ll 1$) and diffusive ($r_{c}\geq 1)$ transport regimes, see Figure 1d. While these have been observed in open systems and are well-characterized for structureless noises [13] and for other related systems [58,59], here, we want to generalize the study to more complex noise scenarios for the integrable XXZ chain.

### 3.1. One Excitation

We now study how varying both our collision parameters, ν and $r_{c}$, affects transport, as shown in Figure 2. In general, as would be the case in a standard Markovian regime, we see that increasing $r_{c}$ towards the diffusive regime decreases the transport speed. As the collisions are more frequent, the transport rate is reduced by the projective nature of the collisions, leading to a transport slowdown, which asymptotically leads to the quantum Zeno [11,13] regime and the freezing of dynamics. By analyzing the magnetization on the central site, see Figure 2a, we can observe that for time-heterogeneous noise (low ν) the loss of coherence is significant and occurs already at tJ∼2. For noise uniformly distributed in time (high ν), instead, the oscillatory behavior highlights that some level of coherence is preserved, especially in the low-noise limit (low $r_{c}$) for the studied times. Thus, we can conclude that the homogeneous regime, closer to the Markovian limit, preserves stronger levels of coherence in the system (for equal $r_{c}$), due to the fact that collisions occur on average in orderly time intervals.

In addition to analyzing the local magnetization profiles, we quantify the delocalization behavior and transport rate making use of the inverse participation ratio (IPR) [38]. The IPR is
(6)$$\mathrm{IPR}=\sum_{i=1}^{N}\langle i|\rho \left(t\right){|i\rangle }^{2},$$
where *i* represents the site index and |i〉 represents the single-excitation localized states $\left\{|i\rangle \;:\;|i\rangle =\sigma_{i}^{†}{|0\rangle }^{⊗N}\right\}$ [33]. Thus, IPR is bounded between the complete delocalization asymptotic value IPR=1/N, and IPR=1, corresponding to complete localization, i.e., when the excitation remains on a particular site of the network [38]. Therefore, the larger the IPR, the more localized the excitation is over the lattice. We generally expect that, in the absence of dissipation, long-lasting Bloch oscillations will set in, while in the long-time limit in the presence of dephasing, the excitation will spread over the lattice and coherence will disappear, tending toward a configuration where the excitation is completely delocalized over the chain. Thus, the rate at which we approach this state—and consequently the evolution of the IPR—strongly depends on the noise parameters.

In Figure 2b, we provide an analysis of the IPR behavior with varied noise parameters, and we identify different regimes. For fixed temporal heterogeneity with ν=0.5, we observe that, in general, increasing $r_{c}$ implies a slowdown in the transport, as this is the case of structureless noise and a quick damping of the oscillatory behavior. Note that the IPR is less sensitive to coherence than the local magnetization, due to the fact that it is a lattice-averaged measure. From the analysis of both magnetization and IPR, we can see that in the large $r_{c}=100$ and low ν regime, we have groups or bunches of ancillas interacting with the system in short intervals of time. In these cases, the effect of subsequent ancillas is small, given that the first has already projected and locally disentangled the system and time is required for the system to rebuild entanglement. Thus, this regime presents a lower effective $r_{c}$ and smaller noise sensitivity. In fact, this can be confirmed if comparing datasets with the same $r_{c}$ but different ν. The case of noise uniformly distributed in time (ν=100), with collisions occurring on average at the same rate in each location, even within individual realizations, has a notably smaller impact on the system coherence. Note how these oscillatory behaviors have a frequency dependence on the value of $r_{c}$, as we will discuss in detail in Figure 3.

If we analyze the results from fixed collision rates $r_{c}$, for low collision rates ($r_{c}∼0.5,1$) we observe a moderate transport slowdown, less evident for smaller shape parameters, as discussed before. With the presence of high-noise, we study up to $r_{c}∼100$, the transport instead suffers a heavy slowdown.This effect is a manifestation of the quantum Zeno effect [11], leading to the freezing of the transport.

We provide a summary of this behavior report in Figure 2c, where we display the IPR at a given intermediate time (tJ=4.5) versus ν and $r_{c}$. There, we observe how (i) the increase in the collisional rate $r_{c}$ leads to a general transport slowdown; (ii) while the shape parameters have a small impact on the transport rate, we observe clearly the change in trend due to the effects at ν≪1, where the effective $r_{c}$ is smaller (leading to smaller values of IPR). We also find that for ν∼5, with the specific value being a function of $r_{c}$, corresponds to the noise with a larger impact on transport, with a slow decrease towards ν≫1. This shows how, by tuning both ν and $r_{c}$, there is a high degree of control of the transport, with more tunability if we consider local-site-independent changes. We leave this possibility to future studies.

In order to understand the coherent behavior observed in the time signals, in Figure 3, we analyze the spectrum via the *fast Fourier transform* (FFT) of both the local magnetization time evolution (Figure 3a) and the IPR time evolution (Figure 3b), to characterize the oscillatory effects. In particular, in our analysis, we subtract the time-heterogeneous (ν=1) from the time-homogeneous (ν=100) signal for the same collisional rate $r_{c}$ and search for the main frequencies of the system, removing the DC component. From this procedure, we can observe coherence-driven oscillatory behaviors in the system, with frequencies that vary with $r_{c}$ and seemingly independent of ν. As we increase $r_{c}$, oscillations are washed out by the lack of coherence, and the strong damping of the dynamics making the main peak, always at frequency $∼r_{c}$, becomes smaller. However, this peak still corresponds to the main frequency, apart from the DC component and some spurious effects at low frequencies.

### 3.2. Multiple Excitations

So far we have analyzed the role of noise in the transport but not yet considered its interplay with other competing physical effects. Thus, we now add the effect of the anisotropy Δ to our study and consider how this nearest-neighbor interaction between spins, which governs the interference effects between several traveling excitations, can affect the transport in the system.

Given the exponential growth of the Hilbert space H, even within restricted magnetization sectors, and our need to average over trajectories, which involve the evolution of the operator ρ—whose dimension is ${\left(\dim H\right)}^{2}$—we reduced the size of the system when studying more than one excitation. The smaller chain also exhibited faster convergence, allowing reducing the required trajectories. Furthermore, given that we do not expect new phenomena to appear for large $r_{c}$ where the Zeno regime dominates, we also restricted the study to moderate collisional rates $r_{c}\leq 5$. Thus, we study the magnetization, its spreading, and related observables for different noise structures after varying the anisotropy.

It is important to note that the IPR is defined in the case of a single excitation. Thus, here, we need to generalize IPR to the case of a generic number of excitations. Several extensions to the IPR have been discussed in the literature; e.g., through the introduction of the generalized inverse participation ratio [60]. However, this quantity is computationally costly and does not take advantage of the structure of our simulation.

Instead, we chose to introduce a new quantity, which we denote *inverse ergodicity ratio* (IER):(7)$$\mathrm{IER}(t)=\sum_{j}\langle j|\rho \left(t\right)|j\rangle ,$$
defined in terms of the multiple-excitation states |j〉 that compose our computational basis. In fact, this can be seen as a Fock basis for the occupation number: $\left\{|j\rangle \;:\;|j\rangle =\sigma_{i_{1}}^{+}\sigma_{i_{2}}^{+}\cdots \sigma_{i_{q-1}}^{+}\sigma_{i_{q}}^{+}{|0\rangle }^{⊗N}\right\}$, where the subscripts $i_{l}$ indicate the sites in which we have the *q* excitations, and the state index *j* runs from 1 to d=dimH.

The IER bounds, now, describe the following limiting behaviors: IER=1/dimH implies that the system is in a superposition of all the possible states in our reduced magnetization sector basis, thus the state is *ergodic* [61]; IER=1 instead refers to the case in which the system is one out of these specific configurations, which forms part of our basis. So, in analogy with the IPR, the states that are more delocalized (low IPR) are also the more ergodic states (low IER).

Focusing on the study of the model with anisotropy Δ, we analyzed all the regimes of noise structures and Δ. We here report on those that manifested interesting properties, as illustrated in Figure 4, for the simpler case of two neighboring excitations. In particular, we show the spreading of magnetization, the evolution of the magnetization, and the IER versus time when varying the two collision parameters $r_{c}$ and ν for two excitations located in the middle of the chain (Figure 4a,b). In contrast with the case of a single excitation, where no regimes of enhanced transport could be found, here in the presence of large anisotropy Δ, increasing collision rates lead to sustained transport, with the noise breaking the pinning of the excitations together, this can be seen as a manifestation of stochastic resonance, which we refer to in the Discussion section. This important result is also supported by the analysis of the case of weaker Δ=2.5, shown in Appendix B, where the effect is still present, though less evident. We also observe some periodic effects appearing at high ν (see the ν=100 case in Figure 4b) with the main frequency component at $r_{c}$.

Since spatial separation of the excitations would drastically reduce their likelihood of interfering, we consider the generality and robustness of these results by initially placing them at a fixed distance (see Figure 4c,d). In Figure 4c, we observe that noise can prevent the certain destructive interference that appears due to coherence in the absence of noise (or small $r_{c}$). In this case, noise enhances the spreading in the central part of the system. This could be useful in engineering irregular architectures beyond linear chains, to prevent interference effects. By analyzing the magnetization and IER in Figure 4d, no important differences are observed by increasing $r_{c}$, apart from a small slowdown of the spreading, in fact regaining a similar behavior of the single excitation case but with a smaller impact from noise. We note that, unlike the case with two neighboring excitations, where the transport was mostly prevented due to the anisotropy, here we observe differences with $r_{c}$. In particular, an increase in the collision rate $r_{c}$ quite rapidly breaks the coherence-driven oscillations, mostly independently of the noise time-homogeneity.

## 4. Discussion

In this work, we have presented a simple framework for the study of transport modulation via dissipation, with relevant applications not only in quantum technologies but also in phenomenological studies for biology. We have explored how systematic tuning over the noise parameters acting on our physical system can modify, in a controllable manner, its transport properties. This was possible by exploiting a recently developed method to describe open quantum systems, based on *stochastic collision models* (SCMs). This description allowed us to tailor the noise, by tuning both the number of collisions occurring over time and their time homogeneity. In so doing, we could access a large variety of bath-induced noise regimes within a sustainable numerical approach. In particular, our analysis of the noisy XXZ chain concluded that, even if structureless noise leads to the unavoidable slowdown of dynamics [13], here it was possible to find regimes where the transport and coherence time can be controlled by the dissipation in a consistent way.

More importantly, when considering several excitations, we found an interesting interplay between collisions and excitation interactions due to anisotropy. From this outcome, we identified regimes in which transport was enhanced when increasing the collision rate. We could also highlight differences in the transport rate within certain parts of the system, when the particles were not initially next to each other. This is a result of the controlled destruction of coherence, which induces destructive interference that hampers the transport in part of the system. All these results contrast with the single excitation or low-anisotropy cases, in which, in general, the higher the presence of noise, the slower the transport rate. In addition, since we observed that noise allows for a quicker spreading in the central part of the system, this mechanism could prove useful when considering more complex lattice topologies. These results, which may appear counterintuitive, can be understood in the context of the long-standing interdisciplinary phenomenon of stochastic resonance [62]. This phenomenon occurs when an increase in random noise causes an improvement in signal transmission or detection performance instead of the expected decrease. This effect is based on system nonlinearities and relies on the appearance of an *optimal level* of noise that screens interference phenomena but does not strongly perturb the system. Stochastic resonance has been observed in microscopic solid-state systems [63,64,65,66,67] and biological systems, such as neuronal systems [68,69], as well as in classical signal processing [70] and in plant media [71].

All in all, our results suggest that it is possible to find so-far unexplored scenarios where noise can assist the system transport. This can be especially useful in the presence of other competing mechanisms that prevent transport or have more complex connectivities. Thus, our results can be relevant in the investigation of disordered systems [72], so far also studied from the one-particle perspective through IPR [73] and in the context of open quantum systems [74], as well as in randomly connected networks [33] relevant to biology problems, where we could enhance and control transport using the engineering of collision models. We believe our study constitutes one of first steps in building this understanding and fostering this exploration.

## Figures and Tables

**Figure 1 entropy-26-00020-f001:**
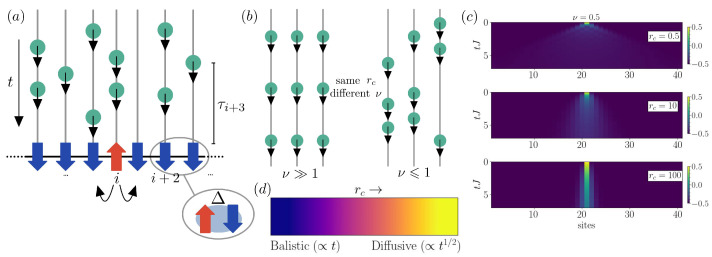
Concept of the model. (**a**) Diagram of an anisotropic spin chain with exchange rate *J* and interaction strength Δ subject to stochastic collisional noise. Over time, individual ancillas (green circles) collide with individual spins in the chain with characteristic times $\tau_{i}$; (**b**) The noise distribution governing the collisions in Equation (Equation 3) can be tuned to be strongly space and time heterogeneous. In particular, on the left-hand side we observe temporal and spatial homogeneity due to the high values of ν, while on the right-hand side, for small values of ν, we have strongly temporal heterogeneity; (**c**) Time evolution of the local magnetization $\langle \sigma_{i}^{z}\left(t\right)\rangle$, depicting the spreading of the initial excitation (spin defect at the central site) as we increase the collision rate $r_{c}=0.5,10,100$ (top to bottom) with a fixed shape parameter ν=100 (time homogeneous noise) for a system with N=41 spins. We observe that, similarly to the case of Markovian dephasing, the spreading velocity decreases with the increasing dissipative rate. (**d**) Phenomenology of transport slowdown due to dephasing between the ballistic regime ($r_{c}\ll 1$), where the system presents linear spreading (∝t) and a diffusive regime ($r_{c}\geq 1)$, leading to slower propagation $\left(\propto t^{1/2}\right)$, both appearing at long times/distances in Markovian dynamics.

**Figure 2 entropy-26-00020-f002:**
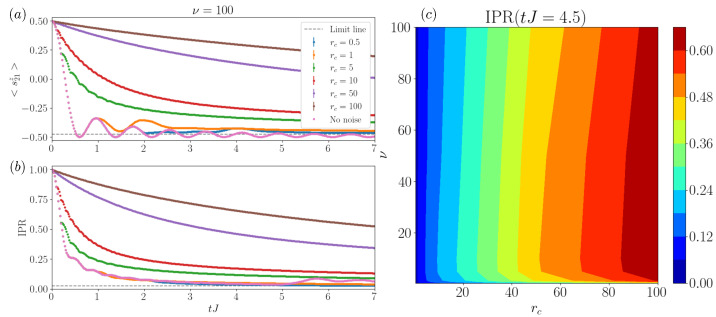
Transport of a single excitation. Magnetization and inverse participation ratio (IPR) evolution in a system with 41 spins and Δ=0. (**a**) Magnetization on the central site $\langle s_{21}^{z}\rangle =\langle \sigma_{21}^{z}/2\rangle$, where the excitation starts, versus time with varying collision parameters, $r_{c}$ and ν. Notice that in general, increasing $r_{c}$ values correspond to decreasing transport speeds. (**b**) IPR versus time with varied collision parameters, $r_{c}$ and ν. We observe a change in the transport rates with increasing $r_{c}$ and also reduced coherence times with smaller ν. (**c**) IPR at time tJ=4.5 versus ν and $r_{c}$. Increasing $r_{c}$ leads to larger IPR (slower transport), apart from the limit at low ν, and we find that the greater effect occurs for shape parameters ν∼5. Note that errorbars are included in the plot but their size is comparable to the marker size. The simulation was performed with dt=0.02 and M=500 samples.

**Figure 3 entropy-26-00020-f003:**
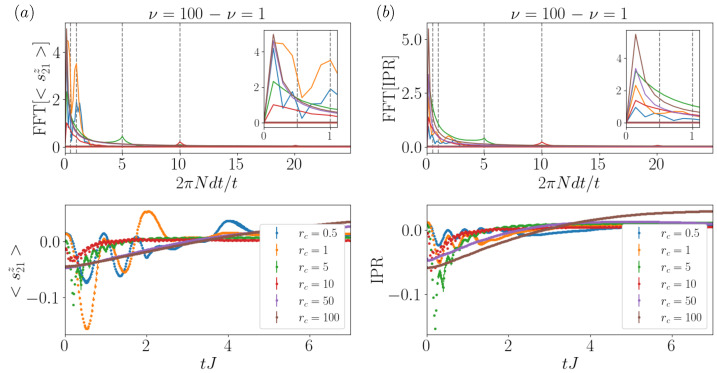
*Fast Fourier transform* analysis for the single excitation case. Both of the time signals analyzed were obtained by subtracting the curves corresponding to ν=100 and ν=1 at equal $r_{c}$, and we have removed the zero frequency component. The results correspond to a system with 41 sites, 1 excitation and Δ=0. (**a**) FFT of the central magnetization signal (top panel) and the signal itself (bottom panel). We observe that the main peak corresponds with the respective $r_{c}$ values, apart from some spurious effects at low frequency. (**b**) FFT of the IPR signal (top panel) and the signal itself (bottom panel). We confirm the presence of the largest peak at $r_{c}$. The insets correspond to both spectra at low frequencies to facilitate visualization.

**Figure 4 entropy-26-00020-f004:**
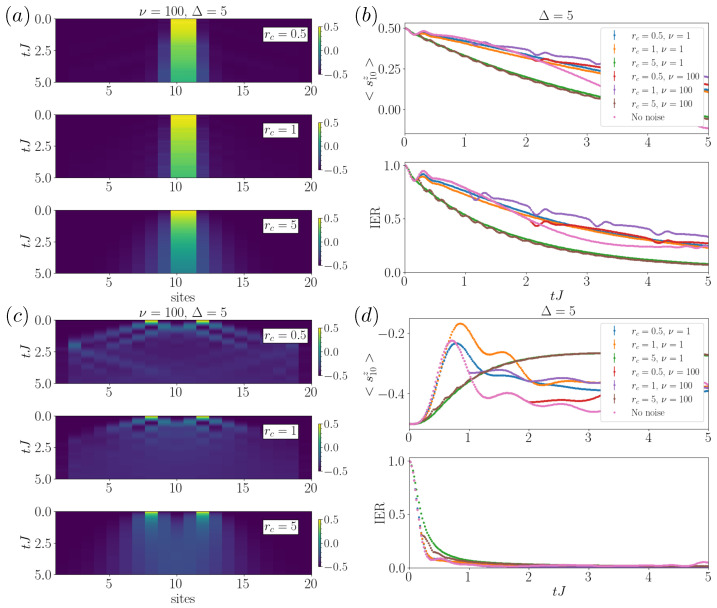
Multiple excitations case. (**a**) Spreading of neighboring excitations with fixed ν=100. Here, we consider an initial state in which the excitations are located in the middle sites, with the magnetization spreading over the chain with different rates, depending on $r_{c}$. (**b**) Evolution of the magnetization (top panel) and of the inverse ergodicity ratio (IER) (bottom panel) versus time for different noise collisional rates $r_{c}$ and shape parameters ν as in the legend for two neighboring excitations. We observe that noise prevents the pinning of the excitations. For the case of homogeneous noise, frequency-dependent modulations appear. (**c**,**d**) are the same as (**a**,**b**), with the two excitations initially separated by three spins. From the transport side, we observe small changes with $r_{c}$. However, we find that noise can prevent coherent destructive interference, enhancing transport in localized regions of the chain. This simulation was performed in a 20-site lattice, with Δ=5, and varying the collision rate $r_{c}=0.5,1,5$. The numerical parameters correspond to dt=0.02 and M=250 samples.

## Data Availability

Data and source codes are available upon request to the authors. Data are contained within the article.

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
