# Peer review of "Engineering Transport via Collisional Noise: A Toolbox for Biology Systems"

_entropy, 2023, doi:10.3390/e26010020_

Round 1

Reviewer 1 Report

Comments and Suggestions for Authors

The authors consider the effect of noise in the transport properties of one and two excitations in a linear XXZ chain. The environmental noise has been described in terms of a stochastic collision model, with the collision statistics being described by a Weibull distribution and the pairwise system-ancilla qubits interaction described by a dephasing conditional rotation. The topic is of timely interest, the manuscript is well written and contain new results. A minor comment: When discussing the transport properties of two excitations along the chain the authors often refer to “interactions” between the excitations. This is not correct: the effects observed by the authors is due to interference between the excitation hopping as two excitations cannot “sit” in the same site.

Comments on the Quality of English Language

There are minor linguistic errors e.g., sensible should be sensitive etc.

Author Response

Dear Referee,

in regards to the Manuscript entropy-2746106, titled “Engineering Transport via Collisional Noise: a Toolbox for Biology Systems”, the authors would like to thank you for your comments that we believe help improve the clarity of the exposition and the discussion of our results. Below we provide a reply to your suggestions. We also provided a pdf with tracked changes in our resubmission for further clarity on the changes performed.

Reply to Referee 1:

Comments and Suggestions for Authors

The authors consider the effect of noise in the transport properties of one and two excitations in a linear XXZ chain. The environmental noise has been described in terms of a stochastic collision model, with the collision statistics being described by a Weibull distribution and the pairwise system-ancilla qubits interaction described by a dephasing conditional rotation. The topic is of timely interest, the manuscript is well written and contain new results. A minor comment: When discussing the transport properties of two excitations along the chain the authors often refer to “interactions” between the excitations. This is not correct: the effects observed by the authors is due to interference between the excitation hopping as two excitations cannot “sit” in the same site.

We thank the referee for this comment as indeed this was an oversimplification of the language in our case. Since the anisotropy can be seen as a nearest-neighbor spin-spin interaction and the anisotropy is indeed what governs the specific interference profiles that we observe. As a result, we had often referred to as interaction effects in the text. We have reduced the use of the term "interaction" in section 3.2 and added the following clarification at the beginning of this section: “Thus, we now add the effect of the anisotropy \Delta to our study and consider how this nearest-neighbour interaction between spins, that governs the interference effects between several travelling excitations, can affect the transport in the system.”

Comments on the Quality of English Language

There are minor linguistic errors e.g., sensible should be sensitive etc.

We have corrected the linguistic errors and typos. For details, see the pdf with tracked changes.

Best regards,

The authors

Reviewer 2 Report

Comments and Suggestions for Authors

This paper studies transport of spin excitations through a Heisenberg spin chain in the presence of a randomly fluctuating environment, using the stochastic collisional model described in Eqns. (3) and (4). The time propagation algorithm used to simulate the magnetization dynamics is clearly described in Section 2.

The core results of this work are (i)  in the case of a single excitation showing the interplay between the time homogeneity of collisions and magnetization spreading, but with an overall dephasing of the quantum signal due to noise, and (ii) in the presence of interactions, identifying a regime where the presence of the noisy environment actually enhances the magnetization transport due to a weakening of the excitation coupling. I recommend publication after the following comments are addressed:

1) The authors find the interesting result that  randomly fluctuating environment can actually enhance the magnetization transport. Could they compare this effect to other instances of stochastic resonance found in the literature? The phenomenon of stochastic resonance has evolved into an interesting subfield of nanoscale transport in its own right, so this would be a useful discussion for readers of this work.

2) The inset figure legends in Figs. 2 (a), 3 (a) and (b), 4 (b) are not clear where they overlap with the data. I suggest moving these legends and making them fully opaque.

3) The Gamma function (I assume it is a Gamma function) in Eq. 4 is not defined or specified as such.

4) In their description of the stochastic unravelling algorithm, the authors specify the collision time S_i = S_i +\tau. Is this \tau identical to the mean collision time \tau_th specified in Eq. 4? If so, the authors should say so and should use the same notation for the two \tau parameters. If not, the authors should specify what this \tau parameter is.

Comments on the Quality of English Language

There are some typos and spelling mistakes in the paper (for instance, 'were the collisions' on line 36 of p.1 should read 'where the collisions') but overall the quality of English is high.

Author Response

Dear Referee,

in regards to the Manuscript entropy-2746106, titled “Engineering Transport via Collisional Noise: a Toolbox for Biology Systems”, the authors would like to thank you for your comments that we believe help improve the clarity of the exposition and the discussion of our results. Below we provide a detailed point-by-point reply to your suggestions. We also provided a pdf in the resubmission with tracked changes for further clarity on the changes performed.

Comments and Suggestions for Authors

This paper studies transport of spin excitations through a Heisenberg spin chain in the presence of a randomly fluctuating environment, using the stochastic collisional model described in Eqns. (3) and (4). The time propagation algorithm used to simulate the magnetization dynamics is clearly described in Section 2.

The core results of this work are (i)  in the case of a single excitation showing the interplay between the time homogeneity of collisions and magnetization spreading, but with an overall dephasing of the quantum signal due to noise, and (ii) in the presence of interactions, identifying a regime where the presence of the noisy environment actually enhances the magnetization transport due to a weakening of the excitation coupling. I recommend publication after the following comments are addressed:

1) The authors find the interesting result that  randomly fluctuating environment can actually enhance the magnetization transport. Could they compare this effect to other instances of stochastic resonance found in the literature? The phenomenon of stochastic resonance has evolved into an interesting subfield of nanoscale transport in its own right, so this would be a useful discussion for readers of this work.

  1. We thank the referee for this useful observation. We addressed this comment adding a paragraph on stochastic resonance with supporting literature in the discussion section (Section 4) and a sentence referring to this discussion in page 8 when discussing Fig. 4.

2) The inset figure legends in Figs. 2 (a), 3 (a) and (b), 4 (b) are not clear where they overlap with the data. I suggest moving these legends and making them fully opaque.

      2) We modified the requested figures according to this suggestion. We reduced slightly the font of the legends, shifted their position and made them opaque.

3) The Gamma function (I assume it is a Gamma function) in Eq. 4 is not defined or specified as such.

        3) We clarified the introduction of the Gamma function in the text after Eq. 4 and we included a brief description of why it appears as a result of an integral from the distribution in Eq. 3.

4) In their description of the stochastic unravelling algorithm, the authors specify the collision time S_i = S_i +\tau. Is this \tau identical to the mean collision time \tau_th specified in Eq. 4? If so, the authors should say so and should use the same notation for the two \tau parameters. If not, the authors should specify what this \tau parameter is.

        4) We addressed this comment about the two parameters which are indeed different. While \tau_th is the mean collision time, the second \tau (not clearly defined before) was intended as a random increment sampled from our distribution. We have included a clarification in the paragraph under Eq. 5: “which will be S_i=S_i+\tau, with \tau a random time sampled from the distribution, note that this would in general differ from its average, given by the mean collision time \tau_{th}.”

Comments on the Quality of English Language

There are some typos and spelling mistakes in the paper (for instance, 'were the collisions' on line 36 of p.1 should read 'where the collisions') but overall the quality of English is high.

We have corrected the linguistic errors and typos. For details, see the pdf with tracked changes.

We thank again the referee for their considerations and useful comments.

Best regards,

The authors

Round 2

Reviewer 2 Report

Comments and Suggestions for Authors

The authors have addressed all of my comments well.

I now recommend this paper for immediate publication.